# Pathophysiological Investigation of Skeletal Deformities of Musculocontractural Ehlers–Danlos Syndrome Using Induced Pluripotent Stem Cells

**DOI:** 10.3390/genes14030730

**Published:** 2023-03-16

**Authors:** Fengming Yue, Takumi Era, Tomomi Yamaguchi, Tomoki Kosho

**Affiliations:** 1Department of Histology and Embryology, Shinshu University School of Medicine, Matsumoto 390-8621, Japan; 2Shinshu University Interdisciplinary Cluster for Cutting Edge Research, Institute for Biomedical Sciences, Matsumoto 390-8621, Japan; 3Department of Cell Modulation, Institute of Molecular Embryology and Genetics, Kumamoto University, Kumamoto 860-0811, Japan; 4Department of Medical Genetics, Shinshu University School of Medicine, Matsumoto 390-8621, Japan; 5Center for Medical Genetics, Shinshu University Hospital, Matsumoto 390-8621, Japan; 6Division of Clinical Sequencing, Shinshu University School of Medicine, Matsumoto 390-8621, Japan; 7Research Center for Supports to Advanced Science, Shinshu University, Matsumoto 390-8621, Japan

**Keywords:** musculocontractural Ehlers–Danlos syndrome, *CHST14* (carbohydrate sulfotransferase 14), skeletal deformities, induced pluripotent stem cells (iPSCs), in vitro assessment, impaired osteogenesis

## Abstract

Musculocontractural Ehlers–Danlos syndrome caused by mutations in the carbohydrate sulfotransferase 14 gene (mcEDS-*CHST14*) is a heritable connective tissue disorder characterized by multiple congenital malformations and progressive connective tissue fragility-related manifestations in the cutaneous, skeletal, cardiovascular, visceral, and ocular systems. Progressive skeletal deformities are among the most frequent and serious complications affecting the quality of life and activities of daily living in patients. After establishing induced pluripotent stem cells (iPSCs) from cultured skin fibroblasts of three patients with mcEDS-*CHST14*, we generated a patient iPSC-based human osteogenesis model and performed an in vitro assessment of the phenotype and pathophysiology of skeletal deformities. Patient-derived iPSCs presented with remarkable downregulation of osteogenic-specific gene expression, less alizarin red staining, and reduced calcium deposition compared with wild-type iPSCs at each stage of osteogenic differentiation, including osteoprogenitor cells, osteoblasts, and osteocytes. These findings indicated that osteogenesis was impaired in mcEDS-*CHST14* iPSCs. Moreover, the decrease in decorin (*DCN*) expression and increase in collagen (*COL12A1*) expression in patient-derived iPSCs elucidated the contribution of *CHST14* dysfunction to skeletal deformities in mcEDS-*CHST14*. In conclusion, this disease-in-a-dish model provides new insight into the pathophysiology of EDS and may have the potential for personalized gene or drug therapy.

## 1. Introduction

Ehlers–Danlos syndrome (EDS) comprises a heterogeneous group of heritable connective tissue disorders affecting as many as 1 in 5000 individuals and is characterized by joint hypermobility, skin hyperextensibility, and tissue fragility [1]. The International Classification of EDS, published in 2017, recognized 13 subtypes according to clinical features and etiologies [2]. Musculocontractural EDS is a recently identified subtype caused by biallelic pathogenic variants in the genes encoding carbohydrate sulfotransferase 14 (mcEDS-*CHST14*: MIM#601776) or dermatan sulfate epimerase (mcEDS-*DSE*: MIM#615539) [3]. To date, 66 patients (48 families) with mcEDS-*CHST14* and 14 patients (eight families) with mcEDS-*DSE* have been described [4,5]. Patients with mcEDS-*CHST14* have distinct craniofacial characteristics, multiple congenital contractures, and progressive multisystem fragility-related manifestations in the cutaneous, skeletal, and ocular systems [4], while those with mcEDS-*DSE* are less common and have fewer characteristics, including joint manifestations and skin features [5].

*CHST14* (MIM*608429) encodes dermatan 4-*O*-sulfotransferase (D4ST1), a regulatory enzyme in the glycosaminoglycan (GAG) biosynthesis pathway that transfers active sulfate to position four of the N-acetyl-D-galactosamine residues of dermatan sulfate (DS) [6,7]. DS, together with chondroitin sulfate (CS) and heparin sulfate, constitute the GAG chains of proteoglycans. DS is implicated in cardiovascular disease, tumorigenesis, infection, wound repair, and fibrosis via DS-containing proteoglycans such as decorin and biglycan [8]. Because decorin plays an important role in the matrix organization and assembly and can bind to type I collagen, the supposed cause of multisystem connective tissue fragility is the impaired assembly of collagen fibrils, resulting from loss of DS in decorin GAG chains [9].

EDS is a connective tissue disorder. Its clinical phenotypes vary significantly, including skin hyperextensibility, joint hypermobility, and widespread tissue fragility, despite sharing the same responsible mutant genes. Low bone-mineral density and deformities frequently occur in certain EDS types (kyphoscoliotic, arthrochalasia, spondylodysplastic, and classical-like EDS), indicating that EDS interferes with bone development and homeostasis [10]. Possible mechanisms include bone abnormalities caused by the genetic defect, less efficient muscle contraction, decreased force transmission from muscle to bone associated with low tendon stiffness, and decreased physical activity due to pain [10]. Spinal deformities are common in patients with mcEDS-*CHST14*. As the deformity progresses, it causes deterioration of trunk balance, respiratory dysfunction, and swallowing disorders, which is one of the most important contributors to lowering the quality of life and activities of daily living in patients [11]. High expression levels of collagen type I and noncollagenous proteoglycans such as decorin and biglycan in bone tissue indicate their important roles during bone mineralization and the development of the osteoid matrix [12]. Therefore, we speculated that bone lesions in mcEDS-*CHST14* patients may be associated with impaired bone mineralization caused by the abnormal organization of collagen networks mediated by decorin and its GAG chains. However, difficulty obtaining tissue samples from patients and limited skeletal phenotypes in mouse models [13,14] hampered our ability to dissect the pathogenesis of skeletal deformities in mcEDS-*CHST14*.

Induced pluripotent stem cells (iPSCs) are somatic cells that have been transcriptionally reprogrammed to an embryonic stem cell (ESC)-like state [15,16]. iPSCs represent a new milestone in our understanding of human development and disease, as well as regenerative medicine, and make it possible to derive all cell types from a patient bearing specific genetic variants, without the ethical concerns of previous approaches. Researchers are now able to use the derivatives of pluripotent stem cells to study disease mechanisms and establish disease models for drug and toxicology testing [17,18].

In the present study, we used patient-specific iPSCs to investigate the entire process of osteogenesis and uncover the pathophysiology of skeletal manifestations in mcEDS-*CHST14*, thereby creating a foundation for the development of etiology-based therapeutic approaches.

## 2. Materials and Methods

### 2.1. Cell Source

Skin samples from three patients with mcEDS-*CHST14* were obtained through Shinshu University Hospital. Patient information is shown in Table 1. All investigations were conducted in accordance with the Declaration of Helsinki principles and were approved by the Ethics Committee of Shinshu University School of Medicine (#610). Written informed consent was obtained from each patient’s parent or guardian. There were no ethical challenges or research limitations on the human iPSCs because they were not derived from embryos.

### 2.2. Generation of iPSC Lines

Skin fibroblasts from each patient were reprogrammed with the Yamanaka factors (*Oct4*, *Sox2*, *Klf4,* and *c-Myc*) packaged in the CytoTune^®^-iPS 1.0 Sendai Reprogramming Kit (Thermo Fisher Scientific, Waltham, MA, USA). After four-factor transduction, mcEDS-*CHST14* fibroblasts were maintained in Dulbecco’s modified Eagle’s medium (DMEM) (Gibco, Grand Island, NY, USA) supplemented with 10% fetal bovine serum (FBS) (Gibco), L-glutamine (Wako, Osaka, Japan), nonessential amino acids (Wako), and penicillin/streptomycin (Wako) for 7–10 days, and then digested and replanted onto mitomycin C-inactivated mouse embryonic fibroblasts (MEFs). Approximately 20 days later, iPSC colonies were mechanically picked and replanted onto MEFs. The cells were incubated in KnockOut™ DMEM/F-12 (Gibco) with 20% knockout-serum replacement (Gibco), 100 mM nonessential amino acids (Wako, Japan), 1 mM sodium pyruvate (Wako), 100 mM 2-mercaptoethanol (Sigma-Aldrich, St. Louis, MO, USA), 50 U/mL penicillin, and 50 mg/mL streptomycin (Wako). Established iPSC lines from the three patients were named A180, A279, and A280. The wild-type (WT) iPSC line 201B7 and 253G1 (Riken Brc, Tsukuba, Japan) was used as a control in this study.

### 2.3. Sanger Sequencing

Sanger sequencing was performed using a BigDye Direct Cycle Sequencing Kit (Thermo Fisher Scientific, Waltham, MA, USA) with M13-tailed PCR primers on an ABI3130xl Genetic Analyzer, in accordance with the manufacturer’s instructions. Primers were designed using primer3 (http://bioinfo.ut.ee/primer3/, accessed on 11 October 2016.) and are shown in Appendix A.

### 2.4. Teratoma Formation

All experimental procedures on animals were conducted in accordance with the Regulations for Animal Experimentation of Shinshu University, following national regulations and guidelines, and were reviewed by the Committee for Animal Experiments and approved by the President of Shinshu University (Approval No. 020008). To confirm the survival and pluripotency of the constructed iPSC lines, 1 × 10^6^ undifferentiated iPSCs were injected intramuscularly under the renal capsules of male mice with severe combined immunodeficiency (NOD mice, 8 weeks of age). Tissues were resected 8 weeks after transplantation, and the samples were prepared for light microscopy and stained with hematoxylin and eosin to confirm teratoma formation.

### 2.5. Osteogenic Differentiation of iPSCs

iPSCs were differentiated into osteogenic lineages using previously published protocols [24]. Briefly, 2 × 10^5^ iPSCs were maintained in feeder-free conditions and differentiation was induced in gelatin-coated 24-well plates with 20% mTeSR1 medium (StemCell Technologies, Vancouver, BC, CA) and 80% osteogenic differentiation medium composed of KnockOut DMEM (Gibco) supplemented with 20% FBS (Gibco), 1% non-essential amino acids (Wako), 0.1 mM 2-mercaptoethanol (Sigma), 2 mM Gluta-MAX (Wako), 10 mM glycerol-2-phosphate (Sigma), 1 nM dexamethasone (Sigma), 50 µg/mL L-ascorbic acid 2-phosphate sesquimagnesium salt hydrate (Sigma), and 1 µM all-trans-retinoic acid (Sigma), and 10 µM rock inhibitor Y-27632 (Wako). On day 2, the medium was replaced with 100% osteogenic differentiation medium without Y-27632 and changed on days 4 and 7.

### 2.6. Immunocytochemistry

The following antibodies were used for immunohistochemistry: Oct4 (1:300; SantaCruz, Dallas, TX, USA), Nanog (1:100; Chemicon, Temecula, CA, USA), SSEA-3 (1:100; Chemicon), SSEA-4 (1:300; Chemicon); Osteocalcin (OCN, 1:100; Thermo Fisher), and Collagen I (ColI, 1:100; SantaCruz). The specificity of these antibodies was validated using appropriate tissues or cells as positive controls. Cells were fixed in 4% paraformaldehyde for 10 min, washed with 0.01 M phosphate-buffered saline, and incubated with primary antibodies at 4 °C overnight. Localization of antigens was visualized with anti-rabbit or anti-mouse immunoglobulin G (IgG) secondary antibodies conjugated with fluorescein (Alexa 568 and 488; Molecular Probes Inc., Eugene, OR, USA). Imaging was performed using a confocal laser scanning microscope (Leica TCS SP8) and a fluorescence microscope (Keyence BZ-X800).

### 2.7. Quantitative Reverse Transcription PCR (qRT-PCR)

Total cellular RNA was prepared using TRIzol reagent (Invitrogen, Carlsbad, CA, USA) according to the manufacturer’s instructions. Genomic DNA was removed using DNase treatment with PrimeScript RT reagent kit with gDNA eraser (Takara Bio, Shiga, Japan). Total RNA (1 ug) was reverse transcribed for single-strand DNA using random primers and reverse transcriptase. qRT-PCR analysis was performed as described previously [25] using the Thermal Cycler Dice Real-Time System (Takara Bio). The mRNA level was normalized to that of beta-actin. The original cycle threshold values were analyzed using the 2^−ΔΔCt^.

### 2.8. Alizarin Red Staining

Alizarin red staining was used to visualize calcium deposition in iPSC-derived osteogenic lineages. Induced iPSCs were fixed in 4% paraformaldehyde for 5 min and washed twice with distilled water. Alizarin red solution (40 mM, pH 4.2; Wako) was applied for 10 min at room temperature. Nonspecific staining was removed by softly rinsing in pure water until only clear water remained. After drying, the samples were immersed in formic acid for 30 min to solubilize the calcium phosphate precipitates. The melted solution was then placed into a 96-well plate and measured at 400 nm. The results were normalized to cell numbers.

### 2.9. Quantification of Calcium Deposition

Induced iPSCs were treated with 1 N hydrochloric acid (HCl) for 24 h. The calcium content in the HCL supernatants was quantified with an o-cresolphthalein complexone (OCPC) kit (N-assay L Ca; Nittobo Medical, Tokyo, Japan) and measured at 660 nm (reference 800 nm) using a microplate spectrophotometer. The calcium content was normalized to cell number.

### 2.10. Statistical Analysis

Data from three dependent experiments are presented as mean ± standard deviation. The statistical significance of qRT-PCR and other assays was evaluated by unpaired double-tailed Student’s *t*-tests, and a *p*-value < 0.05 was considered significant.

## 3. Results

### 3.1. Generation of iPSCs from Skin Fibroblasts of mcEDS-CHST14 Patients

Skin fibroblasts of three patients with mcEDS-*CHST14* from the Shinshu University Hospital were harvested for reprogramming. Yamanaka’s four factors were used to reprogram the fibroblasts into iPSCs. Similar to the WT iPSC line 201B7, the iPSC lines established from the three patients (A180, A279, A280) all exhibited the special morphology of pluripotent stem cells, including tightly packed colonies and a high nucleus-to-cytoplasm ratio (Figure 1A). Pluripotency of iPSC clones was confirmed by immunocytochemistry staining for pluripotent markers Oct4, Nanog, SSEA-3, and SSEA-4 in vitro (Figure 1B) and teratoma formation in vivo (Figure 1C). Cells or tissues derived from all three germ layers were found in WT and patient iPSC-derived teratomas.

### 3.2. Identification of CHST14 Gene Mutation

Using Sanger sequencing, we confirmed that all three established iPSC lines inherited the *CHST14* mutations from skin fibroblasts. WT iPSC lines (201B7) were found to have no mutation in the *CHST14* gene.

### 3.3. Sequential Differentiation of mcEDS-CHST14 iPSCs into the Osteogenic Lineage

WT and mcEDS-*CHST14* iPSCs were differentiated into the osteogenic lineage using established protocols [17], which mimic their natural development in bone. The expression patterns of several marker genes of different stages of osteogenic differentiation were monitored by qRT-PCR. *OCT4* is highly expressed in pluripotent cells and becomes silenced upon differentiation [26]. On Day 2, the expression of undifferentiation marker *OCT4* decreased significantly in WT iPSCs and iPSCs of two patients (A180 and A279), but not in A280 cells (Figure 2A). *RUNX2* (*RUNX* family transcription factor 2), which encodes a critical regulator of osteogenic development, was used to identify osteogenic progenitors derived from ESCs [27,28]. The expression level of *RUNX2* during osteogenesis was significantly higher in WT-derived iPSCs than in patient-derived iPSCs (Figure 2B). This prompted the possibility of delayed osteogenic differentiation in iPSCs derived from mcEDS-*CHST14* patients. Osteocalcin (*OCN*) is secreted solely by osteoblasts and plays important roles in metabolic regulation and osteogenesis promotion [29]. On Day 7, the expression of *OCN* was upregulated in all the iPSCs groups, although the level of upregulation was lower in A180- and A279-derived iPSCs compared with that in WT-derived iPSCs (Figure 2C). No significant difference in *OCN* expression was found between WT- and A280-derived iPSCs. Expression of *PHEX* (Phosphate-regulating endopeptidase X-linked), which is a gene responsible for X-linked hypophosphatemia, occurs primarily in osteoblast lineage cells [30]. As the induction of differentiation proceeded, a significant increase in *PHEX* expression was observed on Day 10 in WT- and A180-, A279-derived iPSCs groups, but not in A280-derived cells (Figure 2D). Furthermore, the expression of WT iPSC-derived osteogenesis was much higher than that in patient-derived iPSCs. The immunostaining patterns of OCN on Day 7 were consistent with those obtained by qRT-PCR analyses. The round shape of most of the differentiated cells was consistent with the morphology of osteoblasts (Figure 2E). These data suggested that iPSCs were successfully induced into osteogenic lineages and demonstrated significant differences in gene expression between the patient group and the control group. Some iPSC lines have different rates of differentiation. In order to rule out this possibility, another normal iPSC line 253G1 was induced into differentiation at the same time. The results were shown in Appendix A. These two normal cell lines showed similar results compared with patient-specific iPSCs.

### 3.4. Impaired Osteogenesis of iPSCs Derived from mcEDS-CHST14 Patients

To assess the effect of *CHST14* mutation on osteogenesis during bone formation, alizarin red staining and calcium deposition analysis were performed. Alizarin-positive staining was clearly observed in all groups from Day 4 and gradually became stronger. However, the staining of patient-derived iPSCs was weaker than that of WT-derived iPSCs throughout the entire course of osteogenic differentiation and was barely detectable in A280-derived cells (Figure 3A). The staining results were quantified by resolving alizarin red dye (Figure 3B). Furthermore, OCPC quantitative analyses clearly demonstrated the decrease in calcium deposition in osteogenic lineages of patient-derived iPSCs (Figure 3C). These results implied that mcEDS-*CHST14* patients may exhibit insufficient calcium deposition or mineralization during bone formation. Using another normal iPSC line 253G1 as a normal control, similar results were obtained (Appendix A).

### 3.5. Expression of Collagen and Decorin

Impaired assembly of collagen fibrils was observed in the skin of mcEDS-*CHST14* patients, with linear CS chains stretching from the outer surface of collagen fibrils to neighboring fibrils. In healthy skin, DS chains present as round and tightly wrapped collagen fibrils [31]. To investigate the distribution of collagen and decorin in osteogenic lineages, their gene expression was evaluated on Day 7 of osteogenic differentiation. Higher expression of *COL12A1* (Figure 4A) and lower expression of *DCN* (Figure 4B) were found in patient-derived iPSCs compared with those in WT-derived iPSCs. Furthermore, the reduction in *DCN* expression was more pronounced in A280 iPSCs than in iPSCs from the other two patients (A180 and A279). Although the distribution of type I collagen (ColI) in immunocytochemical staining profiles showed no significant between-group differences on Day 7, decorin staining was stronger in WT-derived iPSCs than in patient-derived iPSCs (Figure 4C). These results elucidated the hypothesized link between reduced decorin expression and insufficient calcium deposition in mcEDS-*CHST14* iPSC-derived osteogenic cells. By comparison with another osteogenic differentiation derived from normal iPSC line 253G1, this insufficient calcium deposition was verified in mcEDS-*CHST14* iPSCs (Appendix A).

Taken together, our findings indicated that *CHST14* mutation led to decreased alizarin staining and calcium mineralization, eventually resulting in impaired osteogenesis in mcEDS-*CHST14* patient-specific iPSCs.

## 4. Discussion

The present study of patient-specific skin fibroblasts reprogrammed into iPSCs and subsequently induced for osteogenic differentiation suggested that the pathogenesis of mcEDS-*CHST14* involved impaired osteogenesis in bone formation. Our sequential induction of osteogenic lineages monitored each differentiation stage of bone formation, including osteoprogenitors, pre-osteoblasts, osteoblasts, and osteocytes. Among the most important findings were the impairments in mineralization and the ability of mcEDS-*CHST14* iPSCs to differentiate into osteoblasts and osteocytes. During osteogenesis, mcEDS-*CHST14* patient-derived iPSCs displayed a lower expression of *RUNX2*, *OCN,* and *PHEX*, and reduced alizarin red staining and calcium deposition than WT-derived iPSCs. Decreased expression of *DCN* and a compensatory increase in the expression of *COL12A1* were also noted in patient-derived iPSCs. These findings suggested that impairment of decorin-mediated assembly of collagen fibrils caused by *CHST14* mutation might be related to the skeletal deformities in mcEDS-*CHST14* patients.

Bone is a dynamic tissue that constantly undergoes formation and resorption. Defects in either process may impair the production of healthy bone. As the central component in the bone matrix, collagen fibers are crucial for the mechanical integrity of resilient bone [32,33]. When a solid trunk cannot be formed because of an abnormality in collagen, bone strength decreases and fractures occur frequently, such as that in osteogenesis imperfecta, a disease caused by mutations in collagen genes [34,35].

Decorin, which is a molecule found exclusively in the extracellular matrix, is expressed in most collagen-rich mesenchymal tissues, such as bone, skin, and blood vessels [36,37]. It binds specifically to the surface of fibrillary collagens via its core protein to regulate collagen fibril formation. Additionally, decorin functions in cell adhesion, migration, proliferation, and signaling [38,39]. In bone tissue, decorin is associated with osteoblast differentiation and matrix deposition, as well as mineralization [40]. The absence of decorin increases the severity of osteogenesis imperfecta [41]. mcEDS-*CHST14* patients are susceptible to the development of scoliosis, thoracolumbar kyphosis, and cervical kyphosis [11]. In our study, the rate of decorin expression was somewhat varied in the different mcEDS-*CHST14* iPSC lines. This may be explained by variations in core protein expression or post-translational events. The site of the genetic mutation of *CHST14* in patient A280 is different from that in patients A180 and A279, whereas the clinical severity is comparable in all three patients. In A280-derived cells, the expression levels of collagen and decorin remained low throughout the entire osteogenic process. Therefore, it is tempting to conclude that insufficient secretion of decorin and a reduced ability of compensatory regulation greatly weakened the calcium deposition in the A280-derived osteogenic lineages. Our recent study revealed that a small amount of DS chains was detectable in cultured skin fibroblasts and urine from patient A280, whereas DS chains are typically undetectable in patients with mcEDS-*CHST14*. Furthermore, the truncated D4ST1 in patient A280 was found to be localized in the cytoplasm and nucleus, whereas the WT D4ST1 was localized in the Golgi apparatus [23]. Type I collagen is a major constituent of bones. Patients with mcEDS-*CHST14* and the model of mcEDS-*CHST14* deficient in *CHST14* (*CHST14^-^*^/-^) presented with misoriented collagen fibrils, disorganized collagen fibers, and decreased skin tensile strength [31,42,43]. Decorin has been proposed to be a possible reason that caused the disturbed assembly of the collagen network [31,43,44]. In this study, the findings of decreased expression of decorin, compensatory increased expression of collagen, and weakened mineralization suggested that an impairment of osteogenesis followed by impairment of decorin-mediated assembly of collagen fibrils could be one explanation for the skeletal deformities in mcEDS-*CHST14* patients, and it is consistent with the clinical research. In a report of 12 mcEDS-*CHST14* patients with spinal deformities, bone histomorphometry studies showed a decreased bone volume with thin trabeculae. Additionally, osteoid volume and osteoid values of the ilium decreased to <50%, indicating decreased bone formation clinically [11].

## 5. Conclusions

The findings of this study highlight the feasibility of using patient-derived iPSCs to uncover skeletal manifestations in mcEDS-*CHST14*. The disease-in-a-dish model is a promising source of personalized drug therapies, especially for rare diseases such as EDS. While this study reveals a novel interaction of decorin defect with bone impairment, the contribution of impaired assembly of collagen fibrils to bone mineralization requires further investigation.

## Figures and Tables

**Figure 1 genes-14-00730-f001:**
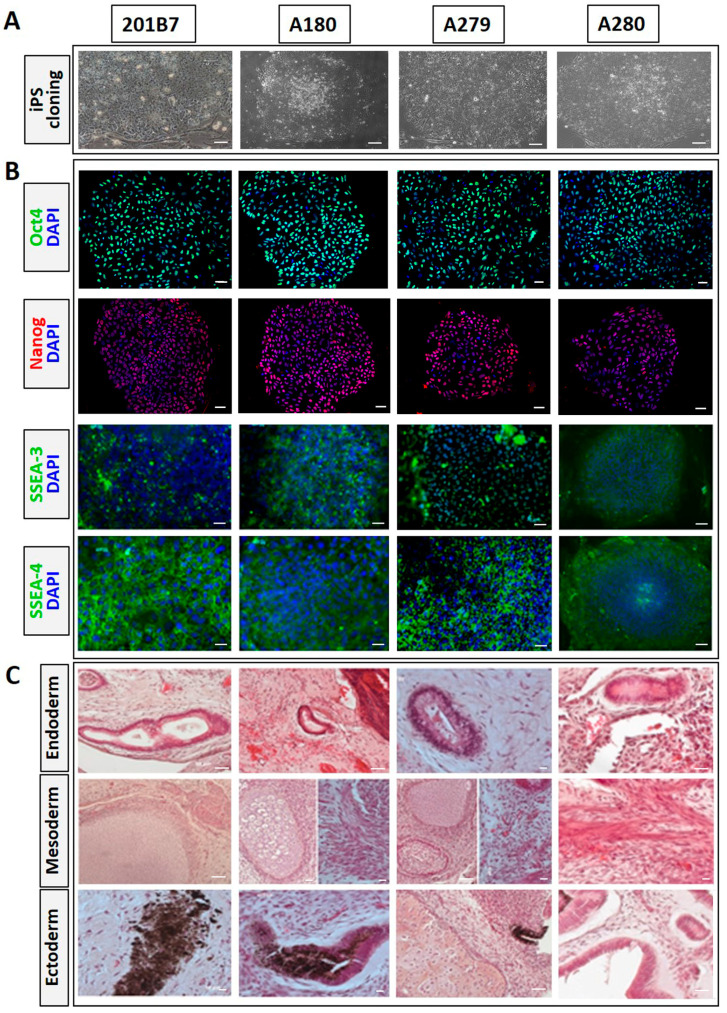
Pluripotency characterization of induced pluripotent stem cells (iPSCs) derived from patients with musculocontractural Ehlers–Danlos syndrome caused by mutations in the carbohydrate sulfotransferase 14 gene (mcEDS-*CHST14*). (**A**) Wild-type (WT) iPSCs (201B7) and iPSCs from three mcEDS-*CHST14* patients (A2:A180, A3:A279, A4:A280) exhibiting tight packing in colonies and a high nucleus-to-cytoplasm ratio. Scale bars, 70 µm. (**B**) Positive staining of all iPSCs for pluripotency markers Oct4, Nanog, SSEA-3, and SSEA-4. Scale bars, 100 µm. (**C**) All iPSC lines were implanted into NOD mice. All three germ-layer structures were found in iPSC-derived teratomas, including gut-tube-like structures (endoderm), smooth muscle or cartilage (mesoderm), and pigmented cells or neural tube (ectoderm). Scale bars, 50 µm.

**Figure 2 genes-14-00730-f002:**
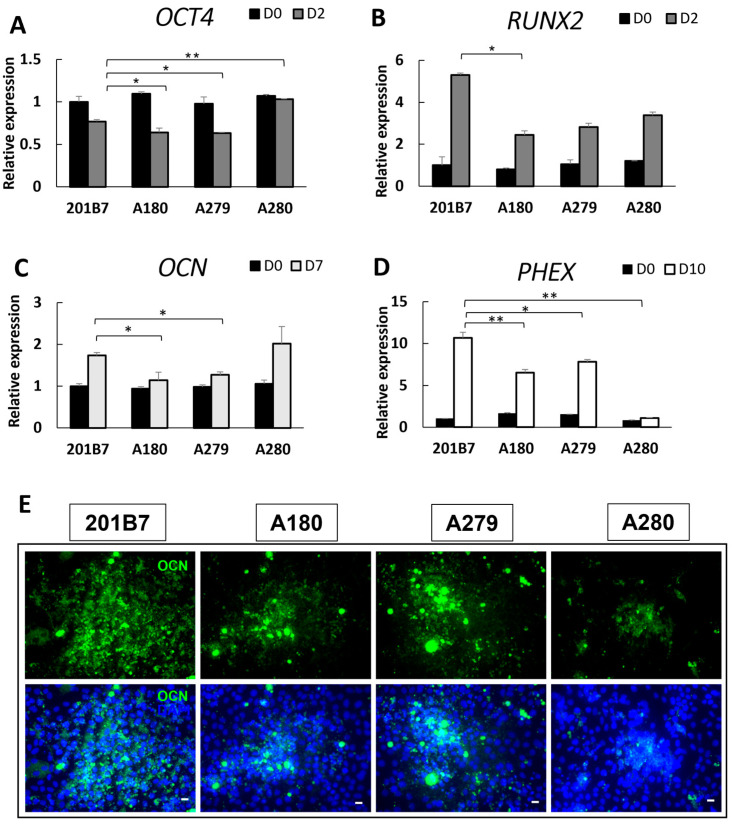
Differentiation of iPSCs into osteogenic lineages. (**A**–**D**) Quantification of the expression of undifferentiated pluripotent stem cell marker *OCT4* and osteogenic differentiation stage-related genes *RUNX2*, *OCN*, and *PHEX* on the indicated day expressed as a value relative to that of WT iPSCs on Day 0. (**E**) Immunostaining of osteoblast-specific protein OCN (green) on Day 7. The presence of the cells was confirmed by staining of nuclei with DAPI (blue). Scale bars, 20 µm. Data expressed as the mean ± SD (*n* = 3); * *p* < 0.05, ** *p* < 0.01.

**Figure 3 genes-14-00730-f003:**
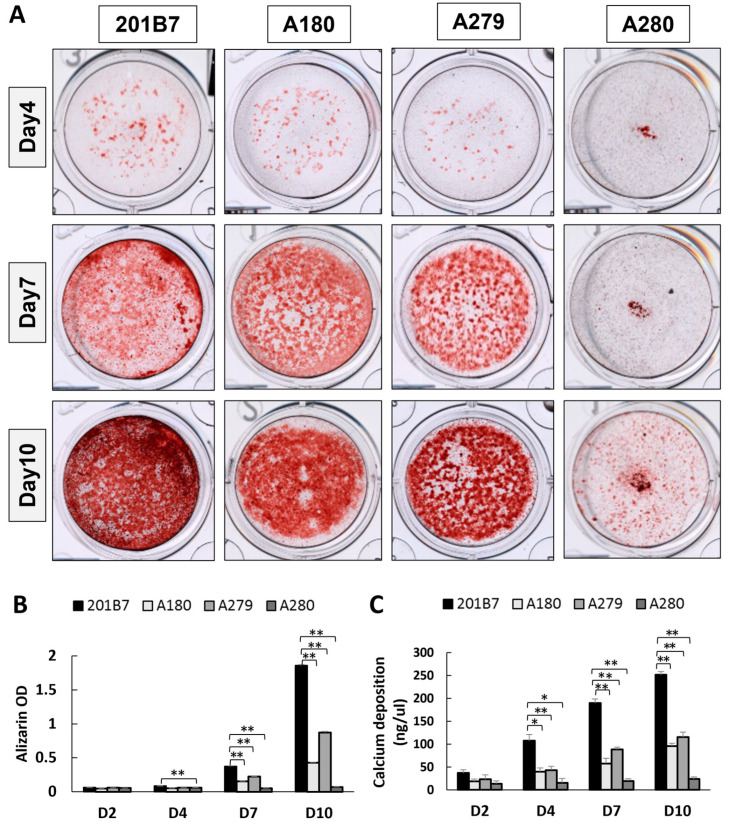
Impaired osteogenesis in mcEDS-*CHST14* iPSCs. (**A**) Representative images of alizarin red staining (red granules) in WT and mcEDS-*CHST14* iPSCs during osteogenic differentiation. Evaluation of bone formation on the indicated day was quantified with dissolved alizarin solution (**B**) and the deposition of calcium (**C**). Results were normalized to cell number. Data expressed as the mean ± SD (*n* = 3); * *p* < 0.05, ** *p* < 0.01.

**Figure 4 genes-14-00730-f004:**
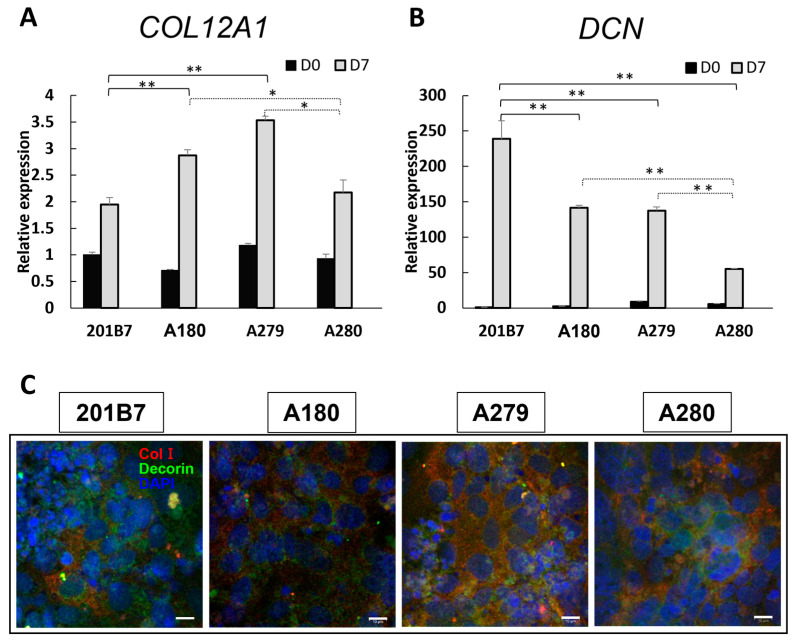
Expression of decorin and collagen in mcEDS-*CHST14* iPSC-derived osteogenic lineages. (**A**, **B**) Gene expression of collagen (*COL12A1*) and decorin (*DCN*) on Day 7, expressed as a value relative to that of WT iPSCs on Day 0. (**C**) Immunostaining of type I collagen (red) and decorin (green) on Day 7. Nuclei were counterstained with DAPI (blue). Scale bars, 10 µm. Data expressed as the mean ± SD (*n* = 3); * *p* < 0.05, ** *p* < 0.01.

**Table 1 genes-14-00730-t001:** Clinical and molecular information of the three patients with musculocontractural EDS (mcEDS)-*CHST14*.

Patient ID	A180	A279	A280
Coding DNA change (NM_130468.4)	c.[842C>T];[878A>G]	c.[842C>T];[878A>G]	c.[2_10del];[2_10del]
Protein alteration (NP_569735.1)	p.[Pro281Leu];[Tyr293Cys]	p.[Pro281Leu];[Tyr293Cys]	p.[?];[?]
Age at skin biopsy	18 y	3 y	12 y
Sex	Female	Male	Male
Ethnicity	Japanese	Japanese	Japanese
Craniofacial features	Typical	Typical	Typical
Congenital contractures	Adducted thumb (bil) Clubfeet (lt)	Adducted thumb (bil) Clubfeet (bil)	Adducted thumb (bil)
Spinal deformity	Kyphoscoliosis (severe, progressive)	Kyphoscoliosis (mild)	Kyphoscoliosis (severe, progressive)
Skin features	Typical	Typical	Typical
Large subcutaneous hematomas	Severe, recurrent	Severe, recurrent	Severe, recurrent
References	Kosho et al., 2005 [19] Kosho et al., 2010 [20] Uehara et al., 2018 [11] Minatogawa et al., 2022 [4] Isobe et al., 2022 [21]	Shimizu et al., 2011 [22] Uehara et al., 2018 [11] Isobe et al., 2022 [21] Minatogawa et al., 2022 [4]	Uehara et al., 2018 [11] Minatogawa et al., 2022 [4] Isobe et al., 2022 [21] Mizumoto et al., 2023 [23]

Bil, bilateral; lt, left; y, years.

## Data Availability

The data that support the findings of this study are available from the corresponding authors upon reasonable request.

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
