# Peer review of "Pathophysiological Investigation of Skeletal Deformities of Musculocontractural Ehlers–Danlos Syndrome Using Induced Pluripotent Stem Cells"

_genes, 2023, doi:10.3390/genes14030730_

Round 1
Reviewer 1 Report
The manuscript by Yue et al. reports study of hiPSCs and their application to Ehlers-Danlos syndrome (EDS) caused by sulfotransferase 14 gene mutations. Specifically, the authors generated 3 novel sets of hiPSCs with 3 unique mutations in this gene. The clonal characterizations included staining of pluripotency markers, sequencing, teratomas, among other assays. The authors then used these lines, plus one normal line, to generate osteogenic cells using standard assays. The differentiated cells were assessed by qPCR, alizarin red staining (a measure of calcium containing osteocytes in differentiated cultures), and calcium deposition quantified with an o-cresolphthalein complexion kit.
Overall, the study as described was well-performed; however, I have a number of concerns. First, this is largely a descriptive study reporting the characterization of three patient derived lines with mcEDS-CHST1 and one normal line. Overall, the characterizations are apt; however, the staining for Nanog could be improved, and in Figure 1B, it is unclear which panel corresponds to SSEA4, as two panels are labelled as SSEA-3. Fro the brightfield image, the colony morphologies of the hiPSC colonies vary considerably, although this may be a function of plating density and time. Line A279, however, does not appear to have a good ratio of nuclei to cytoplasm. Moreover, the immunofluorescent images do not have scale bars, and it appears that some of the magnifications differ among the panels. This makes it difficult to adequately assess. In figure 2, the qPCR data seem apt; however, I have concerns regarding the time points assessed. Generally, the authors only assessed two time points for each gene transcript; however, some hiPSC lines have different rates of differentiation - even with "normal" pluripotency markers. The relative changes, while possibly indicative of the differentiation program, may also be due to altered rates of differentiation among lines. Moreover, all analyses are compared versus a single "normal" line which is not an isogenic control. The addition of another "normal" line and a broader analysis would seem merited to better confirm the reported findings. In Figure 2E, the staining of OCN and DAPI also seems to be highly heterogeneous, and as I look at the data, one of the diseased lines seems highly comparable to the control. Perhaps it is the images presented, but this data seem inconsistent with the RNA data in some instances. This is important, as changes in RNA may not be truly indicative of OCN, etc production. The data in Figure 3 are more convincing. Generally, I do see differences in alizarin red staining, but again, comparisons are to one "normal" line, and only one of the diseased lines shows very dramatic differences. While this may be due to the mutation, it may also be indicative of other problems with this one line. Either an isogenic control or addition of more "normals" is needed. In panels B and C, more of a time course is provided. This is very good, and it is very interesting that A280 does not seem to show increased deposition of either Alizarin red or calcium. This seems to be consistent with Phex qPCR data, which either suggests a direct finding related to this mutation or due to inconsistent/partial differentiation to the osteogenic lineage. Further clarification of these possibilities are needed.
Reviewer 2 Report
This is an elegant study of transformed cells from 3 patients with clinical findings very compatible with musculocontractural EDS who had changes in the associated CHST gene. The system developed to evaluate osteogenesis by the correlated stain Alizarin red gives very compelling results, the speculation about decorin involvement reasonable and appropriately qualified for its basis on one patient. I suspect that similar cell preparations from patients with other types of EDS would show similar deficits in osteogenesis, particularly those with COL1 mutations that give an EDS profile rather than that of osteogenesis, but that is material for future research using this nicely defined system.
Round 2
Reviewer 1 Report
The revised version of this manuscript is improved. In the response to my comments, the authors indicate that experiments with a second control line (normal) were performed and that the data were consistent/similar to that of the one presented throughout the manuscript. However, these data are not shown. I would strongly recommend that data from this additional normal line be included either in the figures of the manuscript (even if only as an inset into the images for the controls lines) but also showing the RNA data. Without seeing this data, it is still difficult to judge some of the potential results.
Round 3
Reviewer 1 Report
The authors have added the images/data to the supplement for a second "normal" line. While some differences were noted with the original description/results relative to the other normal, this addition is very helpful and more convincing.
Thank you for addressing my concerns.